# T4 Lung Carcinoma with Infiltration of the Thoracic Aorta: Indication and Surgical Procedure

**DOI:** 10.3390/cancers15194847

**Published:** 2023-10-04

**Authors:** Marc Irqsusi, Tamer Ghazy, Sebastian Vogt, Nikolas Mirow, Andreas Kirschbaum

**Affiliations:** 1Department of Cardiac Surgery and Thoracic Vascular Surgery, University Hospital Gießen and Marburg (UKGM), 35043 Marburg, Germany; irqsusi@med.uni-marburg.de (M.I.); tamer.ghazy@med.uni-marburg.de (T.G.); sebastian.vogt@med.uni-marburg.de (S.V.); nikolas.mirow@t-online.de (N.M.); 2Department of Visceral-, Thoracic- and Vascular Surgery, University Hospital Gießen and Marburg (UKGM), 35043 Marburg, Germany

**Keywords:** Aorta, aortic stent, cardiopulmonary bypass, infiltration types, lung carcinoma, T4 lung carcinoma

## Abstract

**Simple Summary:**

T4 lung carcinomas with aortic infiltration are rare tumors that pose a real challenge for the treatment team. Therefore, all available findings of the patient should be discussed in an interdisciplinary tumor board in order to coordinate the best possible procedure. In recent years, an increasing number of aortic stents have been implanted prior to tumor resection with good success.

**Abstract:**

Lung carcinomas infiltrate the aorta mostly on the left side and are altogether rare. As an initial step, complete staging is performed and the results are evaluated in an interdisciplinary tumor board. If the patient’s general condition including cardiopulmonary reserves is sufficient, and if there is neither distant metastasis nor an N2 situation, surgical resection may be indicated. The option for neoadjuvant chemotherapy should always be taken into consideration. Depending on the anatomic tumor location, partial lung resection and resection of the affected aortic wall are performed employing a cardiopulmonary bypass. The resected aortic wall is replaced by a vascular prosthesis. In recent years, this proven procedure has partly been replaced by an alternative one, avoiding extracorporeal circulation. An endoaortic stent is implanted in the affected area followed by partial lung resection and resection of the diseased aortic wall. This new procedure has significantly reduced perioperative mortality and morbidity. With proper patient selection, long-term survival can be improved even in this complex malignoma.

## 1. Introduction

Lung carcinomas with infiltration of the aorta are classified according to the current TNM classification [1,2] as T4 tumors. These tumors are characterized by either a size larger than 7 cm or infiltration of adjacent organs such as the diaphragm, mediastinum, heart, trachea, esophagus, spine, or large vessels. Their extent in the thoracic cavity may vary. A T4 tumor is also present when tumor nodules are detected in a lobe other than the primary tumor. Based on their classification, it is understandable that we are dealing with a very heterogeneous group of lung carcinomas. Considering all lung carcinomas, T4 tumors are rare overall, and their percentage is estimated at 4.4% [3]. Lung carcinomas infiltrating the aorta are even rarer, accounting for less than 1% of all T4 tumors [4]. Due to this situation, clinical experience is rather limited; small collectives have been reported in the past, in which a statistical comparison was difficult. There are no randomized studies on the diagnosis and treatment of T4 lung carcinomas with aortic infiltration, leaving clinicians with no evidence-based guidelines. Often the local situation of the patient seems more palliative than curative. In most cases, mediastinal lymph node metastasis is already present, leaving only radiochemotherapy as an option. As a consequence, such patients would not be sent to the surgeon to discuss possible surgical options. Therefore, the treatment of pulmonary carcinomas with aortic infiltration poses a real challenge, especially for the surgeons. This article attempts to describe current aspects in diagnostics and new developments regarding the surgical strategy.

## 2. Diagnostics

### 2.1. Usual Staging Examinations in Lung Carcinoma

Quite often, patients are symptomless, and the lung carcinoma is an accidental finding. The diagnostic starting point here is a computer tomography of the chest with the application of a contrast medium, whereby the spatial proximity of a mass to the aorta suggests an infiltration. As with any lung tumor, location, size and relationship to neighboring structures are analyzed. An FDG PET-CT scan examination and a magnetic resonance imaging of the skull are carried out on each affected person to detect remote metastases. If remote metastasis is suspected, a biopsy of the region in question should be performed. In any case, a bronchoscopy is performed with the intention of a histological confirmation of the tumor, as well as an endobronchial ultrasound examination (EBUS) with lymph node puncture of the relevant lymph node stations. It is essential here to demonstrate or rule out a lymphogenic metastasis, particularly if the lymph nodes are PET-positive. If there are discrepancies between the EBUS and PET-CT scan, a videomediastinoscopic biopsy of the mediastinal lymph nodes should be performed. When these so-called staging examinations are complete, the findings can be assigned to a clinical stage of the disease.

### 2.2. Diagnostic Clues to Aortic Infiltration

Due to the proximity of the lung carcinoma to the aorta, an infiltration can generally be suspected, but not proven. There are local findings of tumors that enclose the aorta almost circumferentially but do not infiltrate it. Depending on the localization, several types are distinguished [5]. In type IA, the main tumor mass is subaortic in the region of the aortic arch. In type IB, the entire aortic arch is affected, usually including the carotids and the left subclavian artery. In type II, there is tumor infiltration below the outlet of the subclavian artery of the thoracic aorta. In type III, there is tumor infiltration in the area of the distal thoracic aorta above the diaphragm (see Figure 1). In local aortic wall infiltration, a distinction can be made between pure adventitial infiltration, full-wall infiltration < 10%, and full-wall infiltration > 10%. A transesophageal echocardiography can be performed to clarify if aortic infiltration is present. A publication by Schroder C. et al. [6] indicates a sensitivity of the method of 91.8% compared with the surgical/pathologic results. In a study by Hong, J.Y. et al. [7], a breath-synchronized magnetic resonance imaging of the chest was carried out. There are reports in the literature [8] that made use of thin-slice CT with 3D reconstruction to clarify an aortic infiltration. Manabe, T. et al. [9] favorize a preoperative dynamic fourth dimension CT (4D–CT) to examine an aortic invasion. In this reported case only the 4D–CT indicated that there was no invasion of the aorta in comparison to a contrast-enhanced computed tomography. Zhao, K. et al. [10] examined 70 patients to compare the diagnostic performance of 3.0-T MRI and CT for aorta invasion in esophageal cancer. They found that the accuracy of 3.0-T MRI is significantly higher than that of CT. Canale, M.L. et al. [11] and Adachi, S. et al. [12] confirmed such results. If the diagnostic situation remains unclear, a diagnostic thoracoscopy is recommended to definitively clarify if aortic infiltration is present [13] and assess the extent of the infiltration in situ. Additionally, there is the possibility to take some probes to prove the tumor infiltration histologically. In case there is no proven infiltration of the aorta by the lung cancer, the operation can continued by doing a video-assisted lung resection in the usual way.

## 3. The Tumor Board: Questions to Be Answered and Recommendations for Surgery

For every patient diagnosed with lung carcinoma, all findings should be discussed in an interdisciplinary tumor board. In accordance with Guirado, M. et al. [14], multidisciplinary care is needed to decide the best therapeutic approach and to provide optimal care to patients with lung cancer. All specialists from thoracic surgery, pneumology, oncology, and radiotherapy relevant to the treatment of lung carcinoma should participate in the tumor board. The first step is to assess the general and nutritional status of the person concerned. In the best case, there is no malnutrition and physical resilience is good. Serious comorbidities such as coronary artery disease, diabetes mellitus, or lung disease should be looked for. Sufficient cardiopulmonary reserves should be available for surgical treatment. To check this point, evaluation of lung function, ergo spirometry, echocardiography, heart load tests, and heart catheter examination in special cases are needed. From an oncological point of view, there should be no remote metastases, merely a T4 situation with localized aortic infiltration. Pneumonectomy should be avoided due to increased morbidity and mortality. Rei, J. et al. [15] found a 9.8% surgery-related mortality after pneumonectomy. The incidence of early complications in 63 pneumonectomies was 35% while 11 patients (17.4%) developed late postoperative complications. In a review of 329 patients having pneumonectomy, there was an overall 30-day mortality rate of 5.1%. Coronary artery disease and respiratory failure were identified as risk factors for mortality [16]. In the most favorable case, lung resection by (sleeve-) lobectomy is possible. If the staging shows an N2 situation, surgery should not be primarily attempted due to the poor prognosis [17,18,19]. If the lung carcinoma is histologically an adenocarcinoma, the prognosis appears better than that of squamous cell carcinoma [20,21,22,23].

In principle, neoadjuvant chemotherapy should be discussed when planning the treatment of T4 lung carcinoma with aortic infiltration [24]. The goal would be to reduce the tumor mass, thereby reducing the extent of the operation, i.e., to be able to remove the lung tumor without an aortic resection [25]. Furthermore, undiscovered micrometastases and tumor cells can be eliminated at the beginning of the treatment.

In cases of initially inoperable patients or borderline findings and a good response to chemotherapy, resection can be considered after the surgeon has re-evaluated the situation.

As a result of the tumor board, the patient is given a recommendation for further treatment. Walter, J. et al. [26] reported that over 90% of recommendations by the tumor board were adherent to the guidelines. If surgery is favored, an interdisciplinary team consisting of cardiac and thoracic surgeons and anesthesiologists should plan the procedure in detail. Questions regarding the best surgical approach, the use of a cardiopulmonary bypass, and the right extent of the removal of the aortic wall should be discussed by the team. The planned surgical strategy should be discussed in detail with the patient regarding its risks and complications.

## 4. Surgical Procedure

The operation usually consists of two parts, with one involving the removal of the lung tumor by a segment resection or lobectomy depending on the extent of the tumor. It is recommended to cover the bronchial stump with a pedicled pericardial flap to prevent graft infection in late bronchial insufficiency. The other part involves the removal of the aortic part of the tumor by means of a subadventitial aortic dissection or a full-thickness resection. In most cases, it can be assumed that the tumor has infiltrated all layers of the aortic wall, necessitating a full-thickness resection. Typically, the adventitia can no longer be separated from the perivascular fatty tissue. Subadventitial aortic dissection is only indicated if a purely superficial tumor infiltration of the aortic adventitia is present. Depending on the location of the tumor, the site may be partially clamped, preserving circulation, or, if extensive resection is required, a heart-lung machine may be needed. The use of a cardiopulmonary bypass should be well planned with the whole team before operation in order to avoid a bad outcome in an emergency situation [25]. If the use of a heart-lung machine is considered to be necessary, there are various ways of establishing it as a classic way. A so-called left heart bypass can be established by cannulating the left atrial appendage using a centrifugal pump and introducing the arterial blood directly into the descending aorta or retrograding into the femoral artery using a centrifugal pump.

Subsequently, there is the possibility of directly clamping out the infiltrated aortic section and replacing it with a homograft, a corresponding vascular prosthesis, or with partial resection with patch plasty using bovine pericardium. The potential negative effects of a left heart bypass as a beating heart concept are to be estimated as lower compared to a classic heart-lung machine. A special focus here is the anesthetic and the monitoring of the patients during the operation.

In addition to large-lumen accesses, it is necessary to measure the arterial pressure in the upper and lower half of the body. Furthermore, it is important to avoid both cardiac arrhythmias, bradycardia and tachycardia, during operation. Hypothermia should also be avoided. Cardiac function can be continuously monitored by transesophageal echocardiography during operation and thus offers a direct possibility to balance the perfusion of the upper and lower half of the body with medication or volume control.

Another access route is the establishment of a heart-lung machine via the arteria and vena femoralis, whereby a long venous cannula is inserted directly into the right atrium under echocardiographic control to relieve the heart. This offers the possibility of a classic perfusion of the lower half of the body as a beating heart model. Arterial monitoring of the upper and lower half of the body is also mandatory here.

### 4.1. Open Operation

The procedure for open surgery depends on the location of the aortic filtration. If type IA is present, a cardiopulmonary bypass with selective perfusion of the brain or deep hypothermia with circulatory arrest is necessary [27,28]. The infiltrated aortic area is replaced with a vascular prosthesis. As an alternative, a fenestrated aortic stent (TEVAR) can be implanted [29], followed by the removal of the tumor-bearing aortic wall down to the prosthesis. Since thoracic stent grafts are primarily blood-tight, it is not necessary to cover the defect after the resection of the aortic wall. However, because an exposed stent graft can trigger a reaction from the surrounding area and lead to considerable tough adhesions with the surrounding tissue, it makes sense to cover the defect with a bovine pericardium, a synthetic Goretex membrane, or a vascular prosthesis cut to size according to the size of the defect.

The procedure for type IB is similar; due to the anatomical location, the tumor can only be removed in deep hypothermia and circulatory arrest. The aortic arch and, if applicable, the supra-aortic branches are replaced with a vascular prosthesis or a homograft (see Figure 2).

If an aortic infiltration type II is present, a normothermic partial bypass (e.g., from the pulmonary or subclavian artery to the distal aorta) of the beating heart is usually sufficient. With an aortic wall infiltration of more than 10% of the aortic circumference, a full-thickness resection with interposition of a Dacron^®^ prosthesis is performed. If the aortic wall infiltration is less than 10% of the aortic circumference, the site is clamped tangentially with a Satinsky clamp and a patch plasty is performed [30]. Maurizi G. et al. [31] described their technique of direct cross-clamping of the aorta and the reconstruction of the aortic arch by direct suturing or polyethylene terephthalate (Dacron) patch. 

In type III, the tumor is located at the thoracoabdominal junction; thus, a combined thoracoabdominal approach should be chosen. In this case, normothermic partial extracorporeal circulation is employed. As with type II, if the aortic wall infiltration exceeds 10%, a partial aortic resection with the placement of an interposed Dacron^®^ prosthesis is performed. If the infiltration is less than 10% of the aortic wall, the aorta can be clamped tangentially with a Satinsky clamp. After the resection of the tumor portion, the defect is treated with a patch [18].

### 4.2. Aortic Stent Implantation

The surgical removal of the tumors in the case of infiltration of the aorta represents a challenge, especially since today’s diagnostic equipment cannot provide 100% accurate information about the depth of the tumor infiltration.

A combined surgical procedure is still associated with increased mortality and morbidity, despite improved surgical techniques and the possibility of intensive medical follow-up care. The increased mortality and perioperative complication rate can be explained using the heart-lung machine, which was used in some case studies.

The use of a heart-lung machine to bridge the cardiovascular function during these operations is often associated with a significantly increased risk of bleeding, a risk of perioperative paraplegia, especially with tumor infiltration of the descending aorta and injury or ischemic triggering of lumbar arteries branching off there, and an increased risk of kidney failure; due to manipulation in the ascending aorta or the aortic arch, there is also an increased risk of stroke. There is also a risk of lung failure or systemic inflammation through the use of the heart-lung machine, which has a significant impact on the outcome. As part of the surgical planning of a stent graft implantation, it is absolutely necessary to carry out a CT angiography of the entire aorta from the supra-aortic vessels to both femoral arteries in advance. This enables adequate planning of the necessary stent graft length and the best access route to implement the intervention. If supra-aortic vessels are involved in the tumor process, so-called fenestrated stent grafts or surgical debranching must be considered. In order to avoid an endoleak, in addition to a minimal oversizing of the stent of up to 10%, an optimal landing zone of at least 3–4 cm in the healthy area is necessary. An optimal coverage of the potential defect area can be seen within the framework of the intervention, which is typically performed in a hybrid operating room using CT angiography, but it makes sense to carry out a control angiography at short intervals of one week in order to rule out a possible peri-interventional endoleak or stent graft migration. Despite the elegant procedure, the implantation of an aortic stent graft also has disadvantages, such as a local inflammatory reaction of the vessel wall as a stimulus to the implanted foreign body within the vascular system. Therefore, some patients can develop a so-called post-implantation syndrome (PIS), which is associated with fever, an increase in leukocytes and the c-reactive protein, increased local signs of inflammation in the sense of edematous expansion of the periaortic tissue, and increased pleural effusions [32]. Since most of the cases involve patients whose operations can be planned electively, a two-stage procedure makes sense. After appropriate planning using the forementioned diagnostic equipment, primarily using an angio-CT scan, the implantation of the aortic stent graft would be carried out first (see Figure 3). After a short interval and corresponding radiological follow-up of an optimal position of the prosthesis, the actual tumor resection can be carried out.

In 2008, Marulli G. [33] described a new indication to endograft to perform resection of the thoracic aorta for infiltration of an adjacent lung cancer into the vessel wall, avoiding a major vascular intervention for aortic graft interposition. Roche-Nagle, G. et al. [34] present two successful cases in 2009 of covered stent grafts deployed in the thoracic aorta to perform resection of the aortic wall infiltrated by a tumor. In 2015, Marulli G. et al. [35] published the results of nine patients in whom an aortic stent was implanted prior to resection due to infiltration of the descending aorta by a lung carcinoma. All patients (six men and three women) were between 52 and 74 years of age. Neoadjuvant chemotherapy was performed in four patients. An aortic stent (length between 100 and 200 mm) was implanted at least 2 to 17 days before the planned operation. A pneumonectomy was performed in four cases and the left lower lobe was removed in five patients. In all cases, the adventitia and media were removed over an average area of 37 by 30 mm. In five cases, the resected area was covered using omentum. Comparable results were reported by Collaud S. et al. [36].

There were also reports about one-stage procedures; after the endovascular stenting the resection of the lung cancer was followed immediately with good results [37].

In 2023, Danial P. et al. [38] described their collective (n = 9) of T4 lung carcinomas with aortic infiltration, with no complication caused by the aortic stent occurring in any case.

Some surgeons have the opinion that thoracic aortic endografting facilitates the resection of tumors infiltrating the aorta [29,39].

Nishioka, N. et al. [40] reported a case where the aortic stent herniated into the chest cavity with bleeding after resection of the aortic wall infiltrated by lung cancer. A re-thoracic endovascular repair was immediately performed. They think that a reinforcement of the aorta through a patch plasty with a bovine pericardium should always be considered. Maybe the risk of infection or fistulization could also be decreased.

## 5. Morbidity and Mortality

The morbidities of the procedures described above were compared as part of a collective study from various clinics [41]. Overall, 11.4% had intraoperative and 25.7% had postoperative complications. The complication rate was lowest in the stent group at 23% (once arrhythmia and twice bleeding). The aorta was clamped tangentially in three patients, and all patients experienced one complication: bleeding in one case and chylothorax in two cases. The complication rate was 100%. Of the six patients who underwent surgery using a cardiopulmonary bypass, 50% experienced complications involving bleeding in two cases and spinal cord injury in one case. All 35 patients treated survived the various interventions. In 2006, Wiebe, K. et al. [42] reported about a collective of 13 patients who underwent an extended pulmonary resection for curative indications requiring support with a cardiopulmonary bypass. The 30-day mortality rate was 15%. Major complications observed were acute lung injury (*n* = 4), right heart failure (*n* = 1), and multi-organ failure (*n* = 1).

Ohta, M. et al. [43] performed a thoracic aorta resection in combination with a lung resection in 16 patients, using a cardiopulmonary bypass in 10 patients. The postoperative mortality rate was 12.5% and the morbidity rate was 31%. In three cases, major intraoperative bleeding occurred. Many authors like Klepetko, W. et al. [19] or de Perrot, M. et al. [27] concluded that a combined resection of the lung and the thoracic aorta with the use of a cardiopulmonary bypass can be performed with an acceptable mortality and morbidity. The key for this is a high selection of the right patients. 

## 6. Long-Term Survival

In their 2017 study, Marulli G. et al. [41] reported the overall survival in all 35 patients of a mean of 31.3 (95% CI: 23.7; 72.9) months. The tumor returned after 22.2 (95% CI: 9.6; 48.1) months. Interestingly, type IA and IB tumors showed a significantly poorer long-term survival compared to type II tumors.

Wex P. et al. [44] compared the long-term survival of a total of 13 patients with T4 lung cancer with aortic infiltration regarding local aortic wall resection. The group with subadventitial resection (*n* = 5) showed a significantly (*p* = 0.001) worse survival (4 vs. 7 months) as compared to those with a full-thickness resection. The authors also showed that the postoperative lymph node status affects 5-year survival (N1: 52% and N2/N3: 39%).

If the operation fails in resecting the lung carcinoma with its aortic infiltration [5] locally in healthy tissue, this has a negative impact on long-term survival [17]. In a study by Fukuse T. et al. [45], 44% of the 15 operated patients were still alive after 3 years after RO resection, but none in the R1 group (*p* = 0.005). Shiraishi, T. et al. [46] reported about their experience with 16 patients with local invasion of the thoracic aorta who underwent resection. The survival of the patients in the complete resection group was found to be 36.5% at 5 years, with two patients surviving more than 5 years without a recurrence.

Long-term survival is also influenced by histological tumor type. Kusumoto, H. et al. [47] showed a better long-term survival (2 years vs. 6 years) for adenocarcinomas (*n* = 8) than for squamous cell carcinomas (*n* = 6).

## 7. Summary

Relative to all lung carcinomas, true aortic infiltration by the tumor is relatively rare. In our opinion, this is a special situation with little evidence on how to handle such tumors. In most cases, the carcinoma grows close to the aorta without invading it. In addition to the usual staging examinations (PET-CT, cranial MRT, and bronchoscopy with EBUS), the diagnostics focus on determining vascular invasion. Despite the high sensitivity of a breath-synchronized magnetic resonance imaging of the chest, the aortic infiltration cannot be verified in every case. In this situation, it is advisable to perform a diagnostic thoracoscopy. It would show us the real situation with the possibility to prove the infiltration by taking biopsies and to continue with the tumor resection in the usual way. After completing the diagnostics, all findings should be discussed in an interdisciplinary tumor board. Affected people in a reduced nutritional and general condition with limited cardiopulmonary reserves are not eligible for surgical treatment. In the case of primarily inoperable or borderline resectable tumors, neoadjuvant chemotherapy with the aim of reducing tumor size can be considered.

If the central mediastinal lymph nodes (N2) are affected, systemic treatment and no surgery should be carried out due to the unfavorable prognosis of the patient [48]. The surgical procedure depends on the location of the aortic infiltration, with three types being distinguished. Tumor infiltrations of the aortic arch (type IA/B) can usually only be removed through open surgery, which is very costly and only utilizing deep hypothermia in circulatory arrest. A complete aortic arch replacement is usually performed with associated morbidity. In recent years, the implantation of an aortic stent before planned resection has increasingly proven its worth. This requires an endoprosthesis made individually for the patient. The prosthesis bridges the tumor-infiltrated part of the aorta, allowing for the resection of the aortic wall down to the prosthesis after the lung tumor has been removed, e.g., by a flap resection. In the following algorithm we give an overview of the surgical strategy in T4 lung cancer invading the aorta (see Figure 4). The morbidity of this procedure is low compared to open procedures using a bypass and should be given preference when technically feasible. The long-term survival of T4 lung carcinomas is influenced by the lymph node status, tumor histology, localization, and resection status. In any case, a complete wall resection of the affected aorta is recommended. According to Muralidaran, A. et al. [49], the use of an unplanned cardiopulmonary bypass seems to be prognostic of unfavorable long-term survival.

T4 tumors with aortic infiltration represent a real challenge for the interdisciplinary treatment team due to their rarity.

However, an experienced interdisciplinary team employing well-designed strategies may improve long-term survival.

## 8. Conclusions

The complex oncological situation in lung carcinomas with infiltration of the aorta can only be improved through well-planned interdisciplinary cooperation. From a surgical point of view, the implantation of an aortic stent should be considered early due to new technical possibilities. Depending on the exact tumor position in relation to the aorta, open resection procedures using a cardiopulmonary bypass are available. The lung is partially excised as is the tumor-infiltrated portion of the aorta. The removed part of the aorta is replaced by a prosthesis. As an alternative, this surgical technique is increasingly replaced by the implantation of an endoaortic stent with subsequent partial lung and aortic wall resection. Presently, there is not yet enough evidence as to whether this procedure should be performed in a single stage or in multiple stages. Certainly, a challenge in future perspective will be stent complications, such as migration or endoleak, and their management.

## Figures and Tables

**Figure 1 cancers-15-04847-f001:**
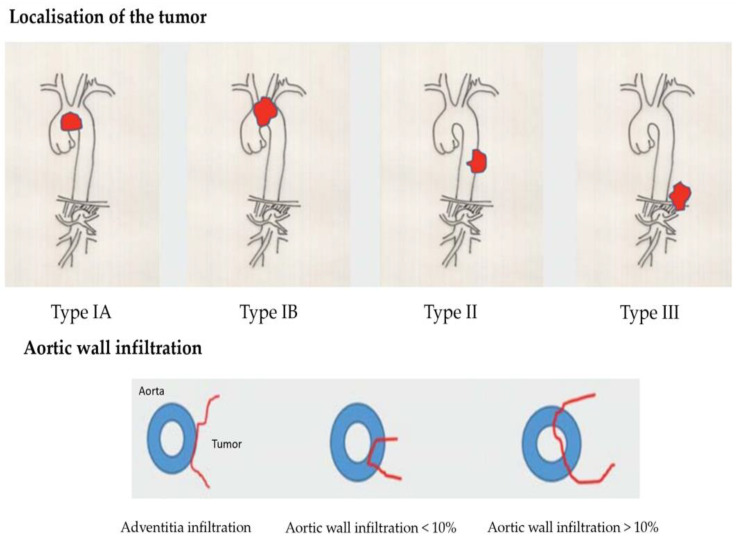
Classification of different types of aortic infiltration caused by lung cancer [5].

**Figure 2 cancers-15-04847-f002:**
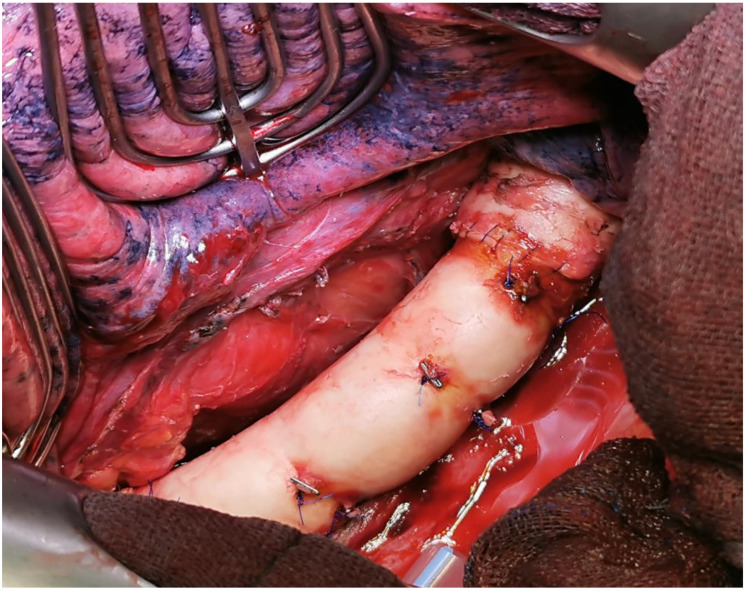
Intraoperative situation after replacement of the descending aorta with a homograft (intraoperative image by permission of A. Rastan, Cardiac Surgery Marburg).

**Figure 3 cancers-15-04847-f003:**
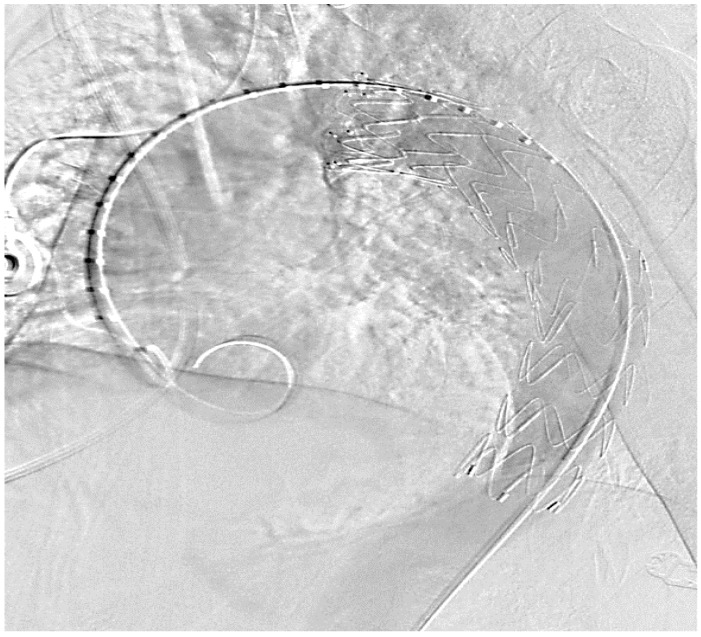
Thoracal Implantation of a covered endovascular stent, control of the correct position (intraoperative picture by M. Irqsusi, Cardiac Surgery, Marburg).

**Figure 4 cancers-15-04847-f004:**
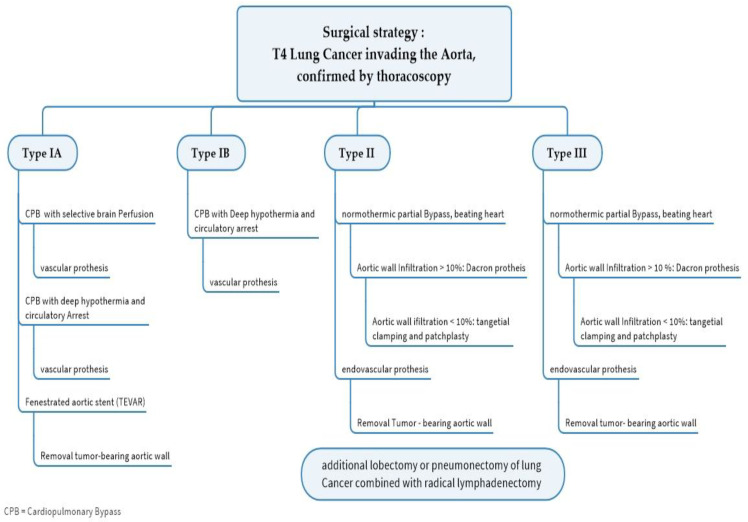
Algorithm for the surgical treatment of T4 lung cancer invading the aorta.

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
