# Peer review of "T4 Lung Carcinoma with Infiltration of the Thoracic Aorta: Indication and Surgical Procedure"

_cancers, 2023, doi:10.3390/cancers15194847_

Round 1

Reviewer 1 Report

Journal of Cancers

Review Article;

The article entitled “T4 lung carcinoma with infiltration of the thoracic aorta: indication and surgical procedure”. To date, only relatively small cohorts of patients with lung carcinomas and aortic infiltration have been reported in the literature. Since no compelling evidence is available in the diagnosis and treatment of such tumors, the aim of the work is to describe essential points that are relevant to indication and surgical procedures and to describe new developments. Despite the increased use of aortic stents and an improved patient selection, the treatment of lung cancer with aortic infiltration still represents a real challenge for the entire treatment team.

Comments for Authors

Ø  The author needs to revise the abstract of the manuscript.

Ø  The author needs to revise the introduction section and explain more about T4 lung carcinoma.

Ø  The author needs to check the effect of the drug taken during the time of treatment.

Ø  Write the keyword in alphabetical order.

Ø  It could be better to describe the questionnaire and responses of the patient sample in the study.

Ø  With “appropriate patient selection” and a well-consid- 335 ered surgical approach, long-term survival can be improved with low morbidity. What does this mean? Revised the sentence.

Author Response

Dear Reviewer 1, 

Thank you for  reviewing our manuscript.

Ø  The author needs to revise the abstract of the manuscript.

 We have created a new abstract, please see below:

New abstract:

      Lung carcinomas infiltrate the aorta mostly on the left side and are altogether rare. As an initial step complete staging is performed and the results are evaluated in an interdisciplinary tumor board. If the patient´s general condition including cardiopulmonary reserves are sufficient, if there is neither distant metastasis nor an N2 situation, surgical resection may be indicated. The option for neoadjuvant chemotherapy should always be taken into consideration. Depending on the anatomic tumor location, partial lung resection and resection of the affected aortic wall is performed employing cardiopulmonary bypass. The resected aortic wall is replaced by a vascular prosthesis. In recent years, this proven procedure has partly been replaced by an alternative avoiding extracorporeal circulation. An endoaortic stent is implanted in the affected area followed by partial lung resection and resection of the diseased aortic wall. This new procedure has significantly reduced perioperative mortality and morbidity.  With proper patient selection, long-term survival can be improved even in this complex malignoma.

Ø  The author needs to revise the introduction section and explain more about T4 lung carcinoma.

We have modified the introduction section

Ø  The author needs to check the effect of the drug taken during the time of treatment.

There were very few data about the effect of induction chemotherapy in T4 non-small cell lung cancer. A study of Lococo F. et.al. (2012) 31 patients received an induction chemotherapy. All patients but one  completed the induction chemotherapy and 27 patients (87%) were operated. A radical resection was possible in 78%. A complete pathologic response was obtained in 22%.[1]

Ø  Write the keyword in alphabetical order.

Corrected in the text

Ø  It could be better to describe the questionnaire and responses of the patient sample in the study.

Thank you for your advice, we think about it

Ø  With “appropriate patient selection” and a well-consid- 335 ered surgical approach, long-term survival can be improved with low morbidity. What does this mean? Revised the sentence.

We have corrected the sentence.

With best regards

Reviewer 2 Report

The present review manuscript titled “T4 lung carcinoma with infiltration of the thoracic aorta: indication and surgical procedure” by Irqsusi et al. is novel and very well organized. The authors covered almost all the aspects as per the rationale of the paper. Overall, the quality of the manuscript is excellent. However, I have 3 major comments as follows.

Comment 1. First of all, the abstract is very poor. The abstract should reflect almost all the aspects briefly and should be discussed in at least 150 to 200 words.

Comment 2. The introduction section is poor. The authors need to provide statistics, challenges in the treatment, challenges in the surgical procedures, advancement, and rationale behind the preparation of this manuscript.

Comment 3. The language of the manuscript is poor. A series of typo mistakes are present in the manuscript.

Comment 4. Conclusion section: I don’t know why the authors have not discussed this section properly. This section is very poor. The authors should make a conclusion on the basis of all the aspects discussed in the manuscript as well as future challenges and possibilities.

Comment 5. References are not organized as per the journal guidelines.

Minor English correction is needed.

Author Response

Dear Reviewer 2,

Thank you for your comments regarding our manuscript:

The present review manuscript titled “T4 lung carcinoma with infiltration of the thoracic aorta: indication and surgical procedure” by Irqsusi et al. is novel and very well organized. The authors covered almost all the aspects as per the rationale of the paper. Overall, the quality of the manuscript is excellent. However, I have 3 major comments as follows.

Comment 1. First of all, the abstract is very poor. The abstract should reflect almost all the aspects briefly and should be discussed in at least 150 to 200 words.

We have created a new abstract, please see below:

New abstract:

      Lung carcinomas infiltrate the aorta mostly on the left side and are altogether rare. As an initial step complete staging is performed and the results are evaluated in an interdisciplinary tumor board. If the patient´s general condition including cardiopulmonary reserves are sufficient, if there is neither distant metastasis nor an N2 situation, surgical resection may be indicated. The option for neoadjuvant chemotherapy should always be taken into consideration. Depending on the anatomic tumor location, partial lung resection and resection of the affected aortic wall is performed employing cardiopulmonary bypass. The resected aortic wall is replaced by a vascular prosthesis. In recent years, this proven procedure has partly been replaced by an alternative avoiding extracorporeal circulation. An endoaortic stent is implanted in the affected area followed by partial lung resection and resection of the diseased aortic wall. This new procedure has significantly reduced perioperative mortality and morbidity.  With proper patient selection, long-term survival can be improved even in this complex malignoma.

Comment 2. The introduction section is poor. The authors need to provide statistics, challenges in the treatment, challenges in the surgical procedures, advancement, and rationale behind the preparation of this manuscript.

Weh ave modified and revised the introduction section.

Comment 3. The language of the manuscript is poor. A series of typo mistakes are present in the manuscript.

The whole text was checked regarding typo mistakes.

Comment 4. Conclusion section: I don’t know why the authors have not discussed this section properly. This section is very poor. The authors should make a conclusion on the basis of all the aspects discussed in the manuscript as well as future challenges and possibilities.

Weh ave changed the conclusion section

Comment 5. References are not organized as per the journal guidelines.

We have formated the references in the style type of MDPI journals with EndNote.

With best regards

Reviewer 3 Report

Comments to Authors:

The authors described current aspects in diagnostics and new developments regarding the surgical strategy for T4–lung carcinomas with aortic infiltration. They suggested that well-planned interdisciplinary collaboration is essential in treatment and that aortic stent placement should be considered early due to new technical possibilities.

This paper is well written.

There are no specific requests regarding the content.

Comment

In Figure 3, the use of upper and lower case letters differs for the initial letters of some words.

Author Response

Dear Reviewer 3,

Thank you for reading and reviewing our manuscript

The authors described current aspects in diagnostics and new developments regarding the surgical strategy for T4–lung carcinomas with aortic infiltration. They suggested that well-planned interdisciplinary collaboration is essential in treatment and that aortic stent placement should be considered early due to new technical possibilities.

This paper is well written.

There are no specific requests regarding the content.

Comment

In Figure 3, the use of upper and lower case letters differs for the initial letters of some words.

We have checked this point.

With best regards

Round 2

Reviewer 2 Report

The revision is satisfactory. I don't have further comments.